# Research Advances in the Analysis of Nitrate Pollution Sources in a Freshwater Environment Using δ^15^N-NO_3_^−^ and δ^18^O-NO_3_^−^

**DOI:** 10.3390/ijerph182211805

**Published:** 2021-11-11

**Authors:** Chao Niu, Tianlun Zhai, Qianqian Zhang, Huiwei Wang, Lele Xiao

**Affiliations:** 1College of Geology and Environment, Xi’an University of Science and Technology, Xi’an 710054, China; niuchao@xust.edu.cn (C.N.); xiaolele@xust.edu.cn (L.X.); 2Institute of Hydrogeology and Environmental Geology, Chinese Academy of Geological Sciences, Shijiazhuang 050061, China; zhaitianlun@mail.cgs.gov.cn

**Keywords:** nitrate, isotope technique, pollution sources, isotopic fractionation, source apportionment

## Abstract

Nitrate is usually the main pollution factor in the river water and groundwater environment because it has the characteristics of stable properties, high solubility and easy migration. In order to ensure the safety of water supply and effectively control nitrate pollution, it is very important to accurately identify the pollution sources of nitrate in freshwater environment. At present, as the most accurate source analysis method, isotope technology is widely used to identify the pollution sources of nitrate in water environment. However, the complexity of nitrate pollution sources and nitrogen migration and transformation in the water environment, coupled with the isotopic fractionation, has changed the nitrogen and oxygen isotopic values of nitrate in the initial water body, resulting in certain limitations in the application of this technology. This review systematically summarized the typical δ^15^N and δ^18^O-NO_3_^−^ ranges of NO_3_^−^ sources, described the progress in the application of isotope technique to identify nitrate pollution sources in water environment, analyzed the application of isotope technique in identifying the migration and transformation of nitrogen in water environment, and introduced the method of quantitative source apportionment. Lastly, we discussed the deficiency of isotope technique in nitrate pollution source identification and described the future development direction of the pollution source apportionment of nitrate in water environment.

## 1. Introduction

Nitrate in the water environment has the characteristics of stable properties, high solubility and easy migration. Therefore, it is usually the main pollution factor in groundwater [1]. In recent years, with the acceleration of urbanization, industrialization, and the rapid growth of population, the impact of human activities (such as excessive application of agricultural chemical fertilizer, substandard discharge of sewage, landfill leakage, chemical fuel leakage, and so on) on the river and groundwater environment has increased, resulting in an increase in the nitrate concentration in freshwater annually. It has become a freshwater environment problem of common concern all over the world and has attracted extensive attention of global scholars [2,3,4].

Nitrate in drinking water is closely related to human health. Studies have found that an excessive nitrate concentration in drinking water can induce a variety of diseases, such as esophageal cancer, gastric cancer, blue baby syndrome, and methemoglobinemia, among other diseases [5,6]. In addition, high concentrations of nitrate in a water environment also have a significant impact on the ecological environment. Research shows that high concentrations of nitrate in surface water can lead to water eutrophication and algal bloom [7], which seriously affect the use value of water body. Therefore, it is particularly important to control nitrate pollution in a freshwater environment.

On the premise of controlling nitrate pollution in water, it is necessary to accurately identify the nitrate pollution sources [8,9]. The traditional method of judging the nitrate pollution sources in a river and groundwater environment is to investigate the land use type of the polluted area and combine the local hydrochemical characteristics [10]. This method is simple, but the results can be inaccurate [11]. In recent years, scholars have found that nitrate from different sources has different nitrogen and oxygen isotope ratios. Therefore, nitrate pollution sources can be better identified according to the δ^15^N and δ^18^O-NO_3_^−^ values [12], which can make up for the shortcomings of traditional methods and provide a means to directly identify pollution sources.

This study summarizes the research progress of identifying nitrate pollution sources in water by stable isotope technology. This paper mainly includes the following aspects: (1) the isotopic value ranges of δ^15^N and δ^18^O of nitrates from different sources are summarized; (2) research progress in identifying the nitrate pollution sources in a freshwater environment by isotope technology; (3) research advances in isotope technology to identify the migration and transformation of nitrogen in freshwater environment; (4) quantitative analysis of nitrate pollution sources by combining with the stable isotope and model in a freshwater environment; and (5) research deficiency and prospect.

## 2. Methods

### 2.1. Search Strategy and Data Sources

The goal of this scoping review is to understand the current advances of pollution sources of nitrate in a freshwater environment. A review protocol was designed in advance to address the research questions, search strategy, data extraction, and analysis. The protocol was not registered.

We reviewed more than 600 published papers searched from Web of Science and China National Knowledge Infrastructure (CNKI) (1990–2021) (based on the keywords “nitrate”, “isotope”, “pollution sources” and “water environment”) and chose 106 of them for this review based on the following criteria: (1) publications are about freshwater (river and groundwater); (2) the δ^15^N-NO_3_^−^ and δ^18^O-NO_3_^−^ values of the potential pollution sources were determined; (3) the contribution proportion of pollution sources was quantified by the stable isotope mixing model; and (4) the pollution sources of nitrate were identified using the multiple isotopes method. The search was not limited by spatial restrictions. The collected literature mainly included articles published from 1990 to 2021. The language of the literature is mainly English or the articles with an English abstract, while papers with the main text written in another language were also included.

### 2.2. Peer-Reviewed Literature Screening

Titles and abstracts of all records were independently screened for eligibility by two authors (C.N. and Q.Q.Z.). Disagreements were solved by consensus or arbitration by a third reviewer (T.L.Z. or H.W.W.).

### 2.3. Data Extraction and Analysis

Data extracted included the following: the δ^15^N-NO_3_^−^ and δ^18^O-NO_3_^−^ values of the potential pollution sources, the kinds of pretreatment technology, and the kinds of isotope and model. Data extraction was conducted by two independent reviewers (C.N. and L.L.X.). The authors (C.N. and T.L.Z.) counted the data of author, year of publication, and study country, and analyzed the δ^15^N-NO_3_^−^ and δ^18^O-NO_3_^−^ values of the potential pollution sources. Disagreements were solved by consensus or arbitrated by a third reviewer (Q.Q.Z. or H.W.W.).

## 3. Determination of Nitrate Isotopes in a Water Environment

The stable isotope ratios for NO_3_^−^ isotopic composition are expressed in delta (δ) units and expressed as a permil (‰) notation relative to an international standard:(1)$$\delta_{\mathrm{Sample}}(‰)=(\frac{R_{sample}}{R_{standard}}-1)\;\times 1000$$
where *R_sample_* and *R_standard_* are the^15^N/^14^N for δ^15^N and ^18^O/^16^O for δ^18^O, respectively. The isotopic values are reported relative to N_2_ in atmospheric air (AIR) and Vienna standard mean ocean water (VSMOW) for δ^15^N and δ^18^O, respectively.

### 3.1. Pretreatment Technology

The pretreatment methods of δ^15^N and δ^18^O of nitrate mainly include the distillation method, the diffusion method, the ion-exchange resin method, bacterial denitrification, and the hydrazoic acid method.

#### 3.1.1. Distillation Method

Distillation is the most traditional method, which includes three steps: (1) reduction of nitrate nitrogen; (2) distillation enrichment; and (3) adsorption of ammonium nitrogen. Firstly, a reducing agent (Devarda) is added to the water sample to reduce nitrate nitrogen to ammonium nitrogen. Then, it is distilled by the Kjeldahl method and adsorbed by acidified filter paper. NH_4_^+^ is converted to N_2_ by means of high-temperature burning, then the nitrogen isotope value is determined by the isotope mass spectrometer [13]. This method is relatively mature, but the sample processing takes a long time, the test work requires special equipment and skilled operators, and it is easy to cause cross contamination.

#### 3.1.2. Diffusion Method

The method involves adding sulfamic acid to the sample to remove nitrite, and then adding Devarda to reduce NO_3_^−^ to NH_4_^+^. Collect the diffused NH_4_^+^ with acid-washed zeolite or acidified suspension, and put the sample into a glass digestion tube for heating and diffusion to convert the collected NH_4_^+^ into N_2_. Finally, nitrogen isotopes are determined by the mass spectrometer [14]. The method is simple and can handle a large number of samples. However, the diffusion period is long (one week or even longer), and incomplete diffusion will cause isotopic fractionation, causing the test results to deviate from the actual value.

#### 3.1.3. Ion-Exchange Resin

The method mainly include four steps: (1) NO_3_^−^ is collected on the ion-exchange columns with an anion resin afterwards and it is eluted from the resins using 15 mL of 3 M HCl. (2) SO_4_^2–^ and PO_4_^3–^ are removed by precipitation with excess BaCl_2_. The sample is then passed through a cation-exchange resin. (3) Excessive Ag_2_O is used to remove Cl^−^ and neutralize the solution to achieve a pH of about 6. (4) The AgNO_3_ solution was freeze-dried for isotopic analysis [7].

This method has little isotope fractionation and high analysis accuracy. However, this method needs to collect a large amount of water samples, the pretreatment procedure is cumbersome, and the treatment cost is high.

#### 3.1.4. Bacterial Denitrification

Bacterial denitrification is a new pretreatment method for simultaneous analysis of nitrogen and oxygen isotopes. In this method, denitrifying bacteria are added to water samples to convert all nitrates into N_2_O. After separation and purification, the generated N_2_O is sent to the isotope ratio mass spectrometer to determine the δ^15^N and δ^18^O value [15,16]. This method requires fewer water samples. At present, it is the mainstream nitrate isotope test method with a simple treatment process, low cost, and high analytical accuracy. However, the bacterial culture cycle is long, and the bacterial culture may be interfered by the toxicity of the samples, causing the test results to deviate from the actual value.

#### 3.1.5. Hydrazoic Acid Method

The treatment process of this method is to add cadmium to the water sample to reduce NO_3_^−^ to NO_2_^−^, and then add azide to reduce NO_2_^−^ to N_2_O. After separation, enrichment, and purification, the generated N_2_O is used to determine the δ^15^N and δ^18^O value [17]. The method has the advantages of a simple pretreatment process, low treatment cost, and fewer water samples, and can realize the automatic injection of a large number of samples. However, the method involves toxic and explosive reagents, which is dangerous during testing. NO_2_^−^ in the water sample will offset the measured δ^15^N and δ^18^O value, and the results need to be corrected.

### 3.2. Isotope Determination Technique

Mass spectrometer is mainly used to determine nitrogen and oxygen isotopes of nitrate in water. With the development of isotope mass spectrometry technology, the joint test of isotope ratio mass spectrometer and element analysis was realized. The element analyzer is connected in series with the mass spectrometer through a continuous flow device. The test process is to wrap the pretreated samples in a silver cup and put them into the automatic sampler of the element analyzer. The subsequent sample pyrolysis, gas injection, and mass spectrometry analysis are completed under the computer control. The test method improves the work efficiency and reduces the sample demand.

## 4. Research Progress in Identifying Nitrate Pollution Sources in a Water Environment by Isotope Technology

Owing to the diversity of nitrate sources in water, as well as physical, chemical, and biological effects, it is impossible to identify the nitrate sources in a water body by traditional measurement of nitrogen content and morphological changes [18]. At present, as an important research means, the theory and method of stable nitrogen isotope (δ^15^N) geochemistry have been widely used in the source identification, migration, and transformation of nitrate in various water environments [19,20,21,22].

The research on nitrate pollution source identification by isotope technology has a history of more than 40 years. In the early stage of the study, owing to the limitation of the technology, scholars can only determine the δ^15^N value of nitrate, and qualitatively identify the main sources of nitrate pollution in water by comparing the δ^15^N values of several potential nitrate pollution sources.

Generally speaking, the characteristic value of δ^15^N-NO_3_^−^ generated from animal excreta or sewage is usually between +7‰ and +20‰ [23,24,25]. The nitrate content in precipitation is low (mostly less than 0‰), and the δ^15^N value is negative owing to volatilization and elution. The δ^15^N value of chemical synthetic nitrate fertilizer is about 0‰ [26]. The range of δ^15^N-NO_3_^−^ value of organic nitrogen in natural soil is −3~+8‰. Domestic sewage has higher δ^15^N-NO_3_^−^values, ranging from +4 to +19‰. The range of δ^15^N-NO_3_^−^ value of organic fertilizer is +5~+25‰ [3,7,27,28].

Although different nitrate sources have different characteristic values of δ^15^N-NO_3_^−^ end elements, there are still many overlapping ranges. In addition, the universality of nitrate sources and the influence of nitrogen biogeochemical processes (such as nitrification and denitrification) will change the composition of δ^15^N-NO_3_^−^. The application of δ^15^N alone in identifying the source of nitrate and the law of migration and transformation is limited.

With continuous progress of research methods, oxygen isotope (δ^18^O) began to be applied to the source analysis in a water body. δ^15^N and δ^18^O isotope methods were used to identify and trace the nitrate source in a water environment and its migration and transformation process [4,29,30,31]. Because different nitrate sources (such as precipitation, nitrate fertilizer, and nitrate from nitrification) have different δ^18^O-NO_3_^−^ isotopic characteristics, generally speaking, the range of δ^18^O-NO_3_^−^ from nitrate nitrogen fertilizer, precipitation, and nitrification is 17~25‰, 25~75‰, and −10~10‰, respectively [11,32,33]. Therefore, δ^18^O-NO_3_^−^ can provide additional information for identifying the sources of nitrate.

Previous studies found that the stable isotope tracing technology of δ^15^N and δ^18^O could provide an effective research basis and means [34,35]. However, the complex nitrate pollution sources and nitrogen cycle process in the freshwater environment lead to isotopic fractionation, which changes the initial δ^15^N and δ^18^O values in nitrate. This has an impact on the accurate identification of nitrate pollution sources [35]. In order to improve the accuracy of pollution source analysis and obtain effective information on the migration and transformation of nitrogen in the water environment, nitrogen and oxygen isotopes can be combined with hydrochemical data or other isotopic data to trace nitrate pollution sources and nitrogen biogeochemical processes in more detail. The researchers found that NO_3_^−^ and boron (B) migrate together in the water body, and boron is not affected by the transformation process. In addition, boron isotopes in chemical fertilizer and sewage have different ratios. Therefore, δ^11^B can be combined with nitrate δ^15^N and δ^18^O to identify the pollution source of nitrate accurately [36]. Furthermore, researchers found that the combined use of δ^15^N-NO_3_^−^ and δ^18^O-NO_3_^−^, and δ^2^H-H_2_O and δ^18^O-H_2_O isotope, could increase the accuracy of source resolution results [37]. In addition, the attenuation of dissolved organic carbon (DOC) and the increase in NO_3_^−^ in the watershed may be closely related to the degradation of organic matter [38]. Therefore, the collaborative study of carbon isotopes can provide more important information in this regard. The δ^2^H and δ^18^O of water can identify the recharge source of groundwater, which can provide evidence for identifying the source of nitrate.

It can be seen that the combination of multiple isotope tracing methods and the above control information is an effective method to reduce the research uncertainty. Because the multi-isotope traceability technology is still in the exploratory stage, the multi-isotope traceability mechanism and the multi-isotope and hydrochemical data joint traceability mechanism need to be deeply studied to ensure the accuracy of nitrate pollution source analysis results in a water environment.

## 5. Definition of Nitrogen and Oxygen Isotope Range of Nitrates from Different Sources

### 5.1. Range of δ^15^N-NO_3_^−^ Values from Different Sources

In recent years, a large number of studies on the application of isotope technology to trace nitrate pollution sources in freshwater environment have emerged, and the nitrogen isotope range of nitrate from different pollution sources changed to a certain extent. Therefore, based on nitrate nitrogen isotope ratios data of different pollution sources in nearly 40 peer-reviewed manuscripts, we reaggregated the range of δ^15^N-NO_3_^−^ from different pollution sources (Figure 1).

The δ^15^N value of atmospheric nitrogen deposition is affected by complex chemical reactions in the atmosphere and various human activities, such as combustion of fossil fuels [72]. Therefore, the δ^15^N value range of atmospheric nitrogen deposition in different countries and regions is wide. It can be seen from Figure 1 that the typical δ^15^N range of atmospheric deposition (NO_3_^−^) is −7.7~+5.8‰, with an average of −0.4‰, and the typical δ^15^N range of atmospheric deposition (NH_4_^+^) is −11.1~+ 2.3‰, with an average of −4.3‰.

The organic nitrogen in manure is mainly urea, which is different from synthetic fertilizer urea. The biggest feature is the strong volatilization of ammonia, which leads to the enrichment of ^15^N in NH_4_^+^ in the solution, and causes the formation of NO_3_^−^ extremely rich ^15^N [73]. The ammonia content in sewage is very high, which is prone to ammonia volatilization, resulting in the enrichment of δ^15^N in NO_3_^−^ from sewage. The typical range of δ^15^N is similar to that from manure. It can be seen from Figure 1 that the typical range of δ^15^N in manure and sewage is +5.9~+22.0‰ and +4.6~+18.4‰, respectively, and the average values are +12.7‰ and +11.4‰, respectively.

Soil organic nitrogen mineralization and nitrification form soil natural nitrate. Therefore, the soil δ^15^N value is mainly affected by soil mineralization and nitrification. In addition, it is also related to soil depth [74], vegetation, climate, and site-specific conditions [64]. From Figure 1, the typical δ^15^N range of soil nitrogen is −3.5~+9.0‰, with an average of 2.2‰.

Chemical fertilizer is the synthesis of N_2_ in the atmosphere through artificial nitrogen fixation, with small nitrogen fixation fractionation [75]. Therefore, the typical ranges of δ^15^N in ammonia nitrogen fertilizer and nitrate nitrogen fertilizer change little, which are −2.7~+2.3‰ and −2.0~+4.0‰, respectively. The average values are 0‰ and +0.3‰, respectively, which is close to 0‰.

### 5.2. Range of the δ^18^O-NO_3_^−^ Value from Different Sources

Compared with δ^15^N-NO_3_^−^, δ^18^O-NO_3_^−^ is more conducive to distinguishing atmospheric nitrogen deposition (the value range of δ^18^O is +52.5~+60.9‰) and soil nitrogen (the value range is +0.8~+5.8‰) [67]. Since then, scholars began to use δ^18^O-NO_3_^−^to identify the pollution source of nitrate in a freshwater environment [59,62,68,69]. Previous studies preliminarily determined the value range of δ^18^O in nitrate from different pollution sources. The value range of δ^18^O in atmospheric deposition is +23~+75‰, the value range of δ^18^O of nitrate fertilizer is +18~+24‰, and the value range of δ^18^O formed by nitrification is −5~+7‰ [56,76]. The author collected the data of nitrate oxygen isotope ratio from different pollution sources in nearly 20 literature sources, and further summarized the range of δ^18^O-NO_3_^−^ from different pollution sources (Figure 2).

According to the δ^18^O values of different nitrate pollution sources, we divide the nitrate pollution sources into three categories: (1) atmospheric deposition; (2) nitrates from nitrification (mainly include soil nitrogen, ammonia nitrogen fertilizer, ammonia in rainfall, and nitrates from sewage and manure); and (3) nitrate nitrogen fertilizer. In Figure 2, 10–90% of the box whisker plots represent the typical range of δ^18^O-NO_3_^−^.

Similar to δ^15^N in atmospheric nitrogen deposition, the δ^18^O value of atmospheric nitrogen deposition is affected by complex chemical reactions in the atmosphere [72], resulting in a wide range of δ^18^O values of atmospheric nitrogen deposition and enrichment of ^18^O compared with oxygen in the atmosphere (δ^18^O is 23.5‰) [57]. The reason for the change in δ^18^O value in atmospheric NO_3_^−^ is not clear, which may be caused by different degrees of isotopic fractionation during the formation of NO_3_^−^ in lightning, incomplete combustion of fossil materials in power plants, automobile exhaust, or photochemical reaction [82]. As can be seen from Figure 2, the typical δ^18^O range of atmospheric nitrogen deposition is +25.0~+75.0‰, and the average value is +54.2‰.

Because one oxygen atom in nitrate comes from oxygen and two oxygen atoms come from water molecules [83], the δ^18^O of soil water is usually negative. As a result, the NO_3_^−^ formed by nitrification in soil is significantly lower in ^18^O than that from the atmosphere, and its δ^18^O value range is −10~+10‰. However, the δ^18^O-NO_3_^−^ values measured in the field are obviously greater than the test value in actual work, and it often further exceeds this range. Scholars have put forward various explanations for this. For example, Wassenaar (1995) believes that the higher value is due to that nitrification occurs in summer and a small amount of denitrification may occur [60]. Kendall et al. (2000) consider that there may be more than one nitrification pathway, and different intermediates have different isotopic components, resulting in a large range of δ^18^O values [84]. From Figure 2, the typical range of δ^18^O-NO_3_^−^ produced by nitrification is +3.5~+16.8‰, and the average value is +10.6‰.

The δ^18^O-NO_3_^−^ value of chemical fertilizer is also relatively enriched, because the oxygen atom mainly comes from atmospheric O_2_ (δ^18^O = +23.5‰). As shown in Figure 2, the typical range of δ^18^O in nitrate nitrogen fertilizer is +18.0~+25.7‰, and the average value is +21.7‰.

## 6. Identification of Nitrogen Migration and Transformation in a Freshwater Environment by Isotope Technology

The migration and transformation of nitrogen in a freshwater environment mainly include assimilation, nitrogen fixation, nitrification, denitrification, and so on. The migration and transformation of nitrogen will lead to the fractionation of nitrate isotopes. Therefore, the principle of isotopic fractionation can be used to identify the process of nitrogen migration and transformation in a freshwater environment. In recent years, researchers have carried out in-depth research on the migration and transformation process of nitrogen in a freshwater environment by jointly using a variety of isotopic methods [85,86,87]. This provides a new research idea for identifying the migration and transformation of nitrogen in a freshwater environment.

In the freshwater environment, if nitrification occurs, the δ^18^O-NO_3_^−^ value will be between −10‰–+10‰ [88]. Theoretically, the δ^18^O-NO_3_^−^ value produced by microbial nitrification can be approximately equal to 1/3 of that in the atmosphere plus 2/3 of that in groundwater [89]. Assuming that the value of δ^18^O -O_2_ in the atmosphere is +23.5‰, combined with the measured value of δ^18^O in water, the theoretical value of δ^18^O-NO_3_^−^ produced by microbial nitrification is calculated. Comparing the theoretical value of δ^18^O-NO_3_^−^ with the measured value, we can judge whether nitrification occurs in water [90]. Fadhullah et al. (2020) carried out research on the nitrogen source, migration, and transformation of Bukit Merah reservoir in Malaysia, and jointly applied δ^15^N and δ^18^O isotopes of nitrate and δ^2^H and δ^18^O isotopes of water to identify that the main migration and transformation process of nitrogen in the reservoir is nitrification and mixing [87].

The identification of nitrogen denitrification in the freshwater environment by the isotope technique has also been studied extensively. Scholars have found that microbial denitrification will increase the δ^15^N and δ^18^O values of nitrate, reduce the concentration of nitrate in water, and cause the ratio of δ^15^N and δ^18^O to fluctuate between 1.3 and 2.1:1 [91,92]. Therefore, the denitrification of nitrogen in the freshwater environment can be identified according to the ratio of δ^15^N and δ^18^O of nitrate. Yi et al. carried out high-frequency monitoring for two tributaries of the Huai River Basin, and jointly applied the δ^15^N and δ^18^O isotopes of nitrate to identify that the important process of nitrogen migration and transformation in the river is denitrification [93]. Li et al. (2020) conducted research on nitrogen pollution in Xiangjiang River. Based on the method of combined application of the δ^15^N and δ^18^O and NO_3_^−^/Cl^−^ ratio diagram of nitrate, they found that denitrification has an important impact on the reduction of nitrate concentration in Xiangjiang River [91].

In addition, some scholars found that the main nitrogen migration and transformation processes in the freshwater environment can be analyzed by calculating the ratio of ∆δ^15^N/∆δ^18^O [94,95]. The calculation formula is as follows:∆δ^15^N: ∆δ^18^O = (δ^15^N − δ^15^N_m_) − (^15^ε/^18^ε) × (δ^18^O − δ^18^O_m_)(2)

In the formula, δ^15^N and δ^18^O represent the δ^15^N and δ^18^O values of nitrate in the upstream input section; δ^15^N_m_ and δ^18^O_m_ represent the δ^15^N and δ^18^O values in the selected site; and ^15^ε/^18^ε is 1. When ∆δ^15^N/∆δ^18^O > 0, this indicates that strong assimilation or denitrification may occur in water, while when ∆δ^15^N/∆δ^18^O ≤ 0, this indicates that strong nitrogen fixation and nitrification may occur in water.

## 7. Quantitative Analysis of Nitrate Pollution Sources in a Freshwater Environment

With the deepening of research, scholars have tried to develop an isotope source analysis model to promote the transition from qualitative research to quantitative research in the identification of nitrate pollution sources. In recent years, researchers have proposed a model based on mass balance to evaluate the contribution rate of different sources to the final sink [96]. They believe that, if there are no more than three nitrate pollution sources in the water body, the contribution rate of each pollution source to nitrate pollution can be quantified by the mixed model based on mass balance in theory. The model can be expressed as follows:δ^15^N = f_1_δ^15^N_1_ + f_2_δ^15^N_2_ + f_3_δ^15^N_3_(3)
δ^18^O = f_1_δ^18^O_1_ + f_2_δ^18^O_2_ + f_3_δ^18^O_3_(4)
1 = f_1_ + f_2_ + f_3_(5)
where δ^15^N and δ^18^O represent the isotope values in the sample and f_1_, f_2_, and f_3_ represent the proportion of contributions from three pollution sources.

Deutsch et al. (2006) identified nitrate pollution sources in Mecklenburg River, Germany, and quantitatively analyzed the contribution rate of each pollution source using this model. The results showed that the contribution rates of irrigation water, groundwater, and atmospheric sedimentation to nitrate pollution in river water are 86%, 11%, and 3%, respectively [63].

However, the researchers found that the mass balance model did not take into account some important uncertain factors. The first uncertain factor is the temporal and spatial variability of nitrate δ^15^N and δ^18^O values, the second uncertainty is isotope fractionation in denitrification reaction, and the third uncertainty is that the model has no solution when the final sink contains multiple pollution sources [97]. On account of this, Parnell et al. (2010) developed a stable isotope mixing model (SIAR) based on R statistical software (https://www.r-project.org/, 1 February 2011). Based on the Dirichlet distribution, the model constructs a logical prior distribution under the Bayesian framework, taking into account the above three uncertainties [98].

The SIAR model can be expressed as follows:(6)$$X_{\mathrm{ij}}=\sum_{k=1}^{k}P_{k}\;(S_{\mathrm{jk}}+C_{\mathrm{jk}})+\varepsilon_{\mathrm{jk}}$$
S_jk_~N(q_jk_, ω^2^_jk_)(7)
C_jk_~ N(λ_jk_, τ^2^_jk_)(8)
(9)$$\varepsilon_{jk}∼N(0,\sigma_{j}^{2})$$
where *X*_ij_ is the δ value of isotope j of mixture i, where i = 1, 2, 3, …, N, j = 1, 2, 3, …, J; *P**_k_* is the proportion of source *k*, which needs to be estimated by the model; S_jk_ is the δ value of isotope j from source *k*, which follows the normal distribution with mean ω_jk_ and variance μ_jk_; *C*_jk_ is the fractionation coefficient of isotope j from source *k*, which follows the normal distribution with mean τ_jk_ and variance λ_jk_; and *ε_jk_* is the residual error, which represents the variance between other mixtures that cannot be quantified, and its mean value is 0 and standard deviation are *σ_j_*.

The SIAR model has broad application prospects. In recent years, it has been applied and achieved good recognition results [99]. Xue et al. (2012) used nitrate δ^15^N and δ^18^O values to identify the main sources of nitrate in surface water, and successfully used the SIAR model to quantitatively study the contribution rate of each pollution source. The results showed that the contribution rate of manure and sewage is the highest, followed by soil nitrogen, nitrate nitrogen, and ammonium nitrogen. The contribution rate of rainfall is the lowest [21]. Xue et al. (2013) applied nitrate δ^15^N, δ^18^O, and δ^11^B values to identify the main sources of nitrate in the surface water of Flanders, Belgium, and quantitatively studied the seasonal differences in the contribution rate of potential pollution sources using the SIAR model. The results showed that manure is the main pollution source of nitrate in water (the contribution rate is 9–85%), and the pollution contribution rate of atmospheric sedimentation is the lowest (0.1–24%) in winter. In summer, the main pollution source of nitrate in water is still manure, but the contribution rate decreases (the contribution rate is 8–59%), and the contribution rate of atmospheric sedimentation increases (the contribution rate is 14–47%) [100].

## 8. Research Deficiency and Prospect

Nitrate pollution has become a worldwide water pollution problem, which has attracted extensive attention of scholars all over the world. In recent years, researchers have applied nitrate δ^15^N and δ^18^O isotope technology to carry out many studies on nitrate pollution sources in a freshwater environment [25,26]. However, the accuracy of traceability results needs to be improved. Through summary and analysis, we believe that there are mainly the following deficiencies. Firstly, the determination of the end-member value of nitrate potential pollution source in previous studies is mostly based on the data in the literature, and the end-member values of potential pollution sources in the study area are not measured, which has an impact on the accuracy of the traceability results. Second, although some studies have applied the multi-isotopes traceability method, it is not clear how the multi-isotopes are coupled and used. Third, previous studies focused on identifying the impact of agricultural non-point source pollution (fertilizer and manure) on groundwater nitrate pollution, and did not explore the contribution of urban non-point source pollution to groundwater nitrate pollution. The fourth is that there are many technical methods for identifying nitrate pollution sources and they are messy, and an effective traceability technical system has not been formed.

The stable isotope method has become an important means to identify the source of nitrate pollution in the water environment. The application prospect is broad, and its development direction deserves more attention. In view of the research status in recent years, the author considers that the following aspects should be investigated in-depth in research in the future. First, because there are some differences in the δ^15^N-NO_3_^−^ and δ^18^O-NO_3_^−^ values of nitrate pollution sources in various regions, more accurate source analysis results are obtained by collecting samples of potential nitrate pollution sources in the study area and by measuring the δ^15^N-NO_3_^−^ and δ^18^O-NO_3_^−^ values of potential nitrate pollution sources before traceability. Second, the complexity of nitrate pollution sources and a polluted environment in the freshwater environment, coupled with the existence of isotope fractionation in the process of nitrogen migration and transformation, makes it difficult to identify nitrate pollution sources using δ^15^N-NO_3_^−^ and δ^18^O-NO_3_^−^ alone. Therefore, the nitrogen and oxygen isotopes of nitrate should be combined with other isotopes (such as B, Cl, Sr, H, O, and so on) to further accurately identify the source, migration, and transformation process of nitrogen, which can make up for the shortcomings and improve the accuracy of traceability [33,101]. The identification of nitrate pollution sources in water by multiple isotope techniques is an important direction in the future. Third, the current research on nitrate pollution source analysis in the freshwater environment mostly focuses on the qualitative research level, and there are relatively few studies on quantifying the contribution rate of each pollution source. Therefore, it is necessary to actively introduce and develop advanced quantitative source analysis models (such as the SIAR model), so as to provide a basis for water environment managers to formulate priority control strategies for nitrate pollution. Therefore, the combination of isotope technology and the mathematical model to quantitatively identify the source of nitrate pollution in water and analyze the contribution rate of the source is the focus of future research.

## 9. Conclusions

Nitrate pollution in freshwater environments is a worldwide water quality problem. In order to prevent the continuous increase in nitrate concentration in freshwater environments, it is necessary to accurately analyze the pollution source of nitrate. At present, the most accurate source analysis method of nitrate in freshwater environments is isotope technology. Because there are certain differences in the stable isotopic composition of nitrate nitrogen and oxygen from different sources, δ^15^N and δ^18^O in nitrate can be used to identify the source of nitrate pollution. However, using δ^15^N-NO_3_^−^ and δ^18^O-NO_3_^−^ can not accurately identify the pollution sources of nitrate owing to the complexity of nitrate pollution sources and nitrogen migration and the transformation process in the freshwater environment, as well as the influence of isotope fractionation. Therefore, it is necessary to use a variety of isotope technologies (such as N, O, B, and Sr isotope) to improve the accuracy of source identification. In recent years, the source analysis model has been applied and developed, which makes it possible to quantitatively analyze the contribution rate of nitrate pollution sources in water. Therefore, the future research direction should be the joint application of multiple isotope technologies to accurately identify the pollution sources of nitrate, and bring the results into the source analysis model to quantitatively analyze the contribution rate of nitrate pollution sources in the freshwater environment, so as to provide a basis for water environment managers to formulate priority control strategies of nitrate pollution.

## Figures and Tables

**Figure 1 ijerph-18-11805-f001:**
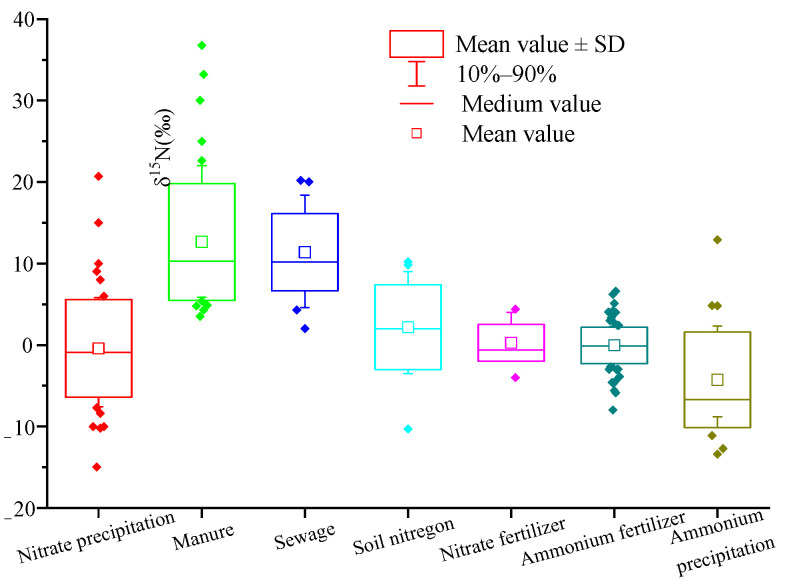
The box plots of δ^15^N values of NO_3_^−^ from different sources. Note: (1) data source: literature [39,40,41,42,43,44,45,46,47,48,49,50,51,52,53,54,55,56,57,58,59,60,61,62,63,64,65,66,67,68,69,70,71]; number of samples: atmospheric deposition NO_3_^−^: *n* = 61; manure: *n* = 58; sewage: *n* = 24; soil nitrogen: *n* = 23; NO_3_^−^ chemical fertilizer: *n* = 18; NH_4_^+^ chemical fertilizer: *n* = 117; atmospheric deposition NH_4_^+^: *n* = 31.

**Figure 2 ijerph-18-11805-f002:**
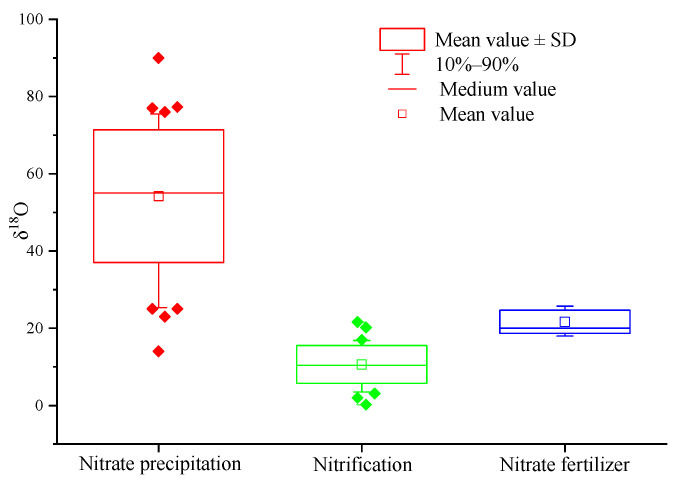
The box plots of δ^18^O values of NO_3_^−^ from different sources. Note: (1) data source: literature [59,60,61,62,63,64,65,66,67,68,69,70,71,77,78,79,80,81], sample number: atmospheric deposition NO_3_^−^: *n* = 40; nitrification: *n* = 36; NO_3_^−^ fertilizer: *n* = 15.

## Data Availability

Not applicable.

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
