# Peer review of "Research Advances in the Analysis of Nitrate Pollution Sources in a Freshwater Environment Using δ15N-NO3 and δ18O-NO3"

_ijerph, 2021, doi:10.3390/ijerph182211805_

Round 1

Reviewer 1 Report

Please see detailed comments attached. Revisions are necessary prior to publication.

Author Response

Overall Comments:

This manuscript is a review of the literature regarding identification of nitrate in water using isotope technology. The main goals of this manuscript are to summarize the typical δ15N and δ18O-NO3 ranges of NO3- sources, describe progress in the application of isotope technique to identify nitrate pollution sources in water environment, analyze the application of isotope techniques in identifying the migration and transformation of nitrogen in water, and introduce the method of quantitative source apportionment. Given the escalating crisis of freshwater shortages globally, it is important to address the contamination of our freshwater sources.

Response: Thank you for your suggestion.

There is no Methods section in this paper or explanation of how the authors conducted this review, such as the databases they searched, or their search terms. Structured literature reviews have processes to follow, and it is not clear what process the authors of this paper followed. This paper should be revised to include a proper Methods section where the authors describe their review process in detail. This should be a requirement in order to allow publication of the manuscript. 

Response: Thank you for your suggestion and we have added the section of method.

The paper needs more finesse, fine tuning, and editing of the style of writing. It delivers the basics of the authors’ research, but there are several areas where sentences should be restructured and reworded in order to flow better. Improving sentence construction in some areas will help the paper to read much better overall. 

Response: Thank you for your suggestion and we have revised the sentence construction.

Specific Comments:

Abstract:

  • The authors start off the abstract (Line 9) by saying “Nitrate is usually the main pollution factor in the water environment”. That is not entirely true. It depends on the type of water (fresh water, groundwater, surface water, ocean, etc.) Authors should clarify.

Response: We have revised “Nitrate is usually the main pollution factor in the water environment” as “Nitrate is usually the main pollution factor in the river water and groundwater environment”.

  • What is meant by “water environment”? Are the authors referring to lakes, ponds,

oceans? The salt water environment is very different from a freshwater environment.

Response: We have revised “water environment” as “freshwater environment” in some paragraphs.

  • Line 21 – “At last” should be “Lastly” or “Finally” .

Response: We have revised “At last” as “Lastly”.

Introduction: 

  • Line 38 – “blue baby disease” is the common term for infant methemoglobinemia, so it should be placed in quotations. Additionally, the correct term is actually “blue baby syndrome”.

Response: We have revised “blue baby disease” as “blue baby syndrome”.

  • Line 39 – remove “etc” and replace with a phrase such as “among other diseases”. Place “and” before methemoglobinemia on line 38.

Response: We have replaced “etc” with “among other diseases” and added the “and” before methemoglobinemia.

  • Line 40 – either say “…..a high concentration of nitrate….” Or “….high concentrations of nitrate…..”

Response: We have revised this sentence.

  • Line 47 – the sentence that ends in “….the local hydrochemical characteristics” requires a citation.

Response: We have added the literature.

  • Line 48 – the word “rough” should be replaced with a more descriptive term. Are the results inconclusive? Are they inaccurate? Please revise.

Response: We have replaced “rough” with “inaccurate”.

  • Line 53-54 – this sentence is awkwardly stated, especially the phrase “On the basis of consulting a large number of literatures” . Please revise this sentence.

Response: We have revised “On the basis of consulting a large number of literatures, the current research progress of identifying nitrate pollution sources in water by stable isotope technology is reviewed.” as “This study summarizes the research progress of identifying nitrate pollution sources in water by stable isotope technology.”.

Research progress in identifying nitrate pollution sources in water environment by isotope technology:

  • Line 70 - Instead of “has a history of more than 40 years”, please use a different phrase such as “has been in existence for more than 40 years” or something similar.

Response: We have rewritten this sentence.

  • Lines 97-102 – this sentences is six lines long, which is much too long. Please restructure this into at least two sentences.

Response: Ok and revised

  • Line 106 – define what “B” is at first use.

Response: We have defined “B” as “Boron”.

  • Line 112 – define what “DOC” is at first use.

Response: We have defined “DOC” as “dissolved organic carbon”.

Definition of nitrogen and oxygen isotope range of nitrates from different sources:

  • Line 127 – Does the term “the author” refer to the author of this paper? That is not the best way to state the work that was done by the authors.

Response: We have revised “The author collected data on nitrate nitrogen isotope ratios of different pollution sources in nearly 40 literatures, and resummarized the range of nitrate nitrogen isotopes of different pollution sources (Fig. 1)” as “Therefore, based on the nitrate nitrogen isotope ratios data of different pollution sources in nearly 40 peer-reviewed manuscripts, we reaggregated the range of nitrate nitrogen isotopes from different pollution sources (Fig. 1).”.

  • Line 127 – 130 – this entire sentence should be revised. Do not say “40 literatures”, say “40 peer-reviewed manuscripts”.

Response: We have revised “40 literatures” as “40 peer-reviewed manuscripts”.

  • Line 144 – remove the parentheses around “combustion of fossil fuels” and say “such as combustion of fossil fuels” instead.

Response: We have modified this place.

Identification of nitrogen migration and transformation in water environment by isotope technology:

  • Line 243 - Replace “studied deeply” with “studied extensively”

Response: We have replaced “studied deeply” with “studied extensively”.

Quantitative analysis of nitrate pollution sources in water environment:

  • Line 287 – citation is needed for the R statistical software

Response: We have added the web site of the R software

  • Line 287 – define “SIAR” or place it in parentheses if it is the abbreviation for the stable isotope mixing model mentioned.

Response: We have rewritten this.

  • Line 302 – Place “The” Infront of “SIAR model”, ie: start off the sentence with “The”.

Response: Ok and revised.

Research deficiency and Prospect:

  • Line 318 – the word “water” used twice in the same sentence. Can remove the term “water environment” here.

Response: Thank you for your reminder, and we have removed the term “water environment” here.

  • Line 323 – the authors should refer to themselves as “we” and not “the author”.

Response: Ok and revised.

  • Lines 322 to 333 is one long sentence. Please break this up into separate sentences based on each point being made, ie: First, second, third and fourth points should all be separate sentences.

Response: Thank you for your suggestion and we have revised the sentence.

Reviewer 2 Report

--------------------
Questions to authors:

Q1: Considering all chemical and biochemical equilibrium between nitrates and other nitrogen sources (nitrification, denitrification, assimilation, etc) for me its hard to believe that we are actually able to "mark" contamination sources by using nitrogen (¹⁵N) and oxygen (¹⁸O) isotopic ratios...  Any comments?

Q2: Using other isotopes (e.g. B, Cl, Sr, H, ...) I believe will not help in resolving this question since they are not "chemically bound" to nitrogen sources and thus free to diffuse and migrate, independently...  Any comments on this?

--------------------
Suggestions to improve:

S1: I believe this review needs a section concerned into instrumental techniques used to access isotopic ratios.

S2: Next section should be dedicated to "standard assumptions" - what are the principles that makes possible to "guess" and "identify" contamination sources?

S3: Following section will be related to numerical evaluation - how to calculate isotopic values...
I believe you are using a relative measure related with "local mean values"?
Otherwise, how to obtain negative values? (e.g. -7.7 o/oo in line 147)

--------------------
To verify:

V1: I believe that in eq.(3) must be a new variable (e.g. gi) instead of fi...  because there is no direct connection to previous variable in eq.(2)?  (due to eventual reactivity of nitrogen sources...)

Author Response

Questions to authors:

Q1: Considering all chemical and biochemical equilibrium between nitrates and other nitrogen sources (nitrification, denitrification, assimilation, etc) for me its hard to believe that we are actually able to "mark" contamination sources by using nitrogen (¹⁵N) and oxygen (¹⁸O) isotopic ratios...  Any comments?

Response: Thank you for your suggestion. Indeed, there are many sources of nitrate pollution (such as sewage, manure, soil nitrogen, rainfall, etc) and the nitrogen cycle process is complex, such as ammoniation, nitrification, denitrification and ammonia volatilization in the water environment. However, previous studies have found that there are some differences of the δ15N and δ18O values from different sources. For example, the δ15N value of sewage, manure and soil nitrogen were + 7~+ 20‰, +5~ +25‰ and -3 ~ +8‰, respectively. The range of δ18O-NO3 from nitrate nitrogen fertilizer, precipitation and nitrification is 17 ~ 25‰, 25‰ ~ 75‰ and - 10‰ ~ 10‰, respectively. Therefore, we can distinguish the sources of nitrate pollution in the water environment by using the nitrogen (¹N) and oxygen (¹O) isotopic ratios. In addition, nitrogen cycle could lead to isotopic fractionation and could change the initial δ15N and δ18O values in nitrate, which has an impact on the accurate identification of nitrate pollution sources. But, we could calculate the enrichment factor of the isotope by using the Rayleigh equation and restore it to the initial isotope value. Therefore, the source of nitrate pollution can be identified by using the nitrogen (¹N) and oxygen (¹O) isotopic ratios.

Q2: Using other isotopes (e.g. B, Cl, Sr, H, ...) I believe will not help in resolving this question since they are not "chemically bound" to nitrogen sources and thus free to diffuse and migrate, independently...  Any comments on this?

Response: Indeed, only using the other isotopes (such as, B) could not identify the pollution sources of nitrate in water. However, we combined the δ11B with δ15N and δ18O could increase the accuracy of source resolution results. This was mainly due to the NO3 and B migrate together in the water body, and B was not affected by the transformation process. When the nitrate pollution source is manure or sewage, only using the δ15N and δ18O of nitrate can not distinguish them accurately due to the δ15N and δ18O ranges of manure (+ 5‰ ~ + 25‰) and sewage (+ 7‰ ~+ 20‰) were overlapped. But, boron isotopes in manure (>+ 10‰) and sewage (<+ 10‰) have different ratios. Therefore, δ11B can be combined with nitrate δ15N and δ18O to identify whether the nitrate comes from sewage or manure.

Suggestions to improve:

S1: I believe this review needs a section concerned into instrumental techniques used to access isotopic ratios.

Response: We have added the section of the instrumental techniques.

S2: Next section should be dedicated to "standard assumptions" - what are the principles that makes possible to "guess" and "identify" contamination sources?

Response: Thank you for your suggestion. The principles of identifying contamination sources of nitrate by the isotope method are to determine the range of isotope values of potential pollution sources. We have defined the value ranges of nitrogen and oxygen of nitrates from different sources in section 5. In addition, the sources of nitrate in water environment can be identified quantitatively by introducing the δ15N and δ18O-NO3 values of the sources and the water samples into the SIAR model

S3: Following section will be related to numerical evaluation - how to calculate isotopic values...
I believe you are using a relative measure related with "local mean values"?
Otherwise, how to obtain negative values? (e.g. -7.7 o/oo in line 147)

Response: We have added the method of calculate isotopic values in section 3.

To verify:

V1: I believe that in eq.(3) must be a new variable (e.g. gi) instead of fi...  because there is no direct connection to previous variable in eq.(2)?  (due to eventual reactivity of nitrogen sources...)

Response: We have revised the formula and reinterpreted the meaning of the letters.

Round 2

Reviewer 1 Report

Overall comments:

Thank you to the authors for addressing most of my previous concerns. The paper has improved.

Thank you for adding a Methods section and describing your literature review process. However, it is much shorter than typical Methods sections in review papers, and I am still not convinced as to the robustness of the review. Is this a systematic review? A scoping review? Was a review protocol created? If so, please report it. What was the search strategy? Was more than one reviewer involved? Typically at least two reviewers should review the articles to be included in the paper. An important criteria of any review paper is that another researcher should be able to recreate the review that was conducted. With the information provided in the newly created Methods section, I do not believe that this is possible. Additional details about the review process should be added before this paper is accepted for publication.

Some review papers have a table that lists all the papers used in the review and other criteria. While this is not completely necessary, I believe that it will add to this paper. See for example table 1 in this paper: Nestler, A. et al. Isotopes for improved management of nitrate pollution in aqueous resources: review of surface water field studies. Environ Sci Pollut Res 18, 519–533 (2011). https://doi.org/10.1007/s11356-010-0422-z

Finally, the authors should take another look at sentence construction throughout the paper. The paper needs more finesse, fine tuning, and editing of the writing style. Some of the writing comes across as though English is not the first language of the authors. This is completely fine, however, the paper may benefit from an English language editor to improve some aspects of the writing style.

Specific comments:

Introduction

  • Line 36 – change “year by year” to “annually”
  • Line 51 – change to “........results can be inaccurate”

Conclusion

  • Line 462, 463 and 465 – change “environment” to “environments”.

Author Response

Reviewer 1:

Overall comments:

Thank you to the authors for addressing most of my previous concerns. The paper has improved.

Thank you for adding a Methods section and describing your literature review process. However, it is much shorter than typical Methods sections in review papers, and I am still not convinced as to the robustness of the review. Is this a systematic review? A scoping review? Was a review protocol created? If so, please report it. What was the search strategy? Was more than one reviewer involved? Typically at least two reviewers should review the articles to be included in the paper. An important criteria of any review paper is that another researcher should be able to recreate the review that was conducted. With the information provided in the newly created Methods section, I do not believe that this is possible. Additional details about the review process should be added before this paper is accepted for publication.

Response: Thank you for your suggestion. We further refined the Methods section of the article. We explained that this article is a scoping review, and supplemented the search strategy. In addition, we added the section of the data extraction and processing procedures.

Some review papers have a table that lists all the papers used in the review and other criteria. While this is not completely necessary, I believe that it will add to this paper. See for example table 1 in this paper: Nestler, A. et al. Isotopes for improved management of nitrate pollution in aqueous resources: review of surface water field studies. Environ Sci Pollut Res 18, 519–533 (2011). https://doi.org/10.1007/s11356-010-0422-z

Response: Thank you for your suggestion. The key data sources in our article have explained in the note of the figure 1 and 2. Thus, this paper did not need a table that lists all the papers used in the review.

Finally, the authors should take another look at sentence construction throughout the paper. The paper needs more finesse, fine tuning, and editing of the writing style. Some of the writing comes across as though English is not the first language of the authors. This is completely fine, however, the paper may benefit from an English language editor to improve some aspects of the writing style.

Response: Thank you for your suggestion. The sentence construction of the paper has been edited by a person with better English writing skills.

Specific comments:

Introduction

Line 36 – change “year by year” to “annually”

Response: We have revised “year by year” as “annually”.

Line 51 – change to “........results can be inaccurate”

Response: We have revised “results are inaccurate” as “results can be inaccurate”.

Conclusion

Line 462, 463 and 465 – change “environment” to “environments”.

Response: We have changed “environment” to “environments” in line Line 462, 463 and 465.

Reviewer 2 Report

I appreciate very much your effort in improving your work.
Thank you for your answers.

Author Response

Reviewer 2:

Comments and Suggestions for Authors:

I appreciate very much your effort in improving your work.

Thank you for your answers.

Response: Thank you for your suggestion.

Round 3

Reviewer 1 Report

Thank you for enhancing the Methods section of the manuscript and adding in additional details about the review process. It is more comprehensive and your methodology is clearer now. Just one minor comment:

  • Line 77 - this sentence is poorly worded. Please restructure this sentence.

Author Response

Comments and Suggestions for Authors

Thank you for enhancing the Methods section of the manuscript and adding in additional details about the review process. It is more comprehensive and your methodology is clearer now. Just one minor comment:

Line 77 - this sentence is poorly worded. Please restructure this sentence.

Response: Thank you for your suggestion. We have revised “The language of the literature is mainly English or the articles with an English abstract, but while the main text written in another language,  were also included.” as “The language of the literature is mainly English or the articles with an English abstract, while the main text written in another language were also included.”.